# Is ChatGPT a Good Causal Reasoner? A Comprehensive Evaluation.

**Jinglong Gao    Xiao Ding*    Bing Qin    Ting Liu**
Research Center for Social Computing and Information Retrieval
Harbin Institute of Technology, China
{jlgao,xding,qinb,tliu}@ir.hit.edu.cn

## Abstract

Causal reasoning ability is crucial for numerous NLP applications. Despite the impressive emerging ability of ChatGPT in various NLP tasks, it is unclear how well ChatGPT performs in causal reasoning. In this paper, we conduct the first comprehensive evaluation of the Chat-GPT's causal reasoning capabilities. Experiments show that ChatGPT is not a good causal reasoner, but a good causal explainer. Besides, ChatGPT has a serious hallucination on causal reasoning, possibly due to the reporting biases between causal and non-causal relationships in natural language, as well as ChatGPT's upgrading processes, such as RLHF. The In-Context Learning (ICL) and Chain-of-Thought (CoT) techniques can further exacerbate such causal hallucination. Additionally, the causal reasoning ability of ChatGPT is sensitive to the words used to express the causal concept in prompts, and close-ended prompts perform better than open-ended prompts. For events in sentences, ChatGPT excels at capturing explicit causality rather than implicit causality, and performs better in sentences with lower event density and smaller lexical distance between events. The code is available on https://github.com/ArrogantL/ChatGPT4CausalReasoning.

## 1   Introduction

Causal reasoning ability is crucial for numerous NLP applications. The recent causal reasoning systems are mainly based on fine-tuned pre-trained language models (PLMs) such as BERT (Devlin et al., 2019) and RoBERTa (Liu et al., 2019). However, their causal reasoning abilities rely on supervised training using large amounts of annotated data.

Most recently, ChatGPT has achieved remarkable performance in various NLP tasks without the need for supervised training. However, there is currently no work that comprehensively evaluates ChatGPT's ability in causal reasoning.

---

*Corresponding Author

Figure 1: The forms of three causal reasoning tasks and the prompts we use. The content that requires ChatGPT to reply is marked in **red**. The multiple-choice CD task also involves samples that ask for selecting the result of the input event. For such samples, we modify the "cause" in the question to "result".

In this paper, we conduct a comprehensive evaluation to demonstrate ChatGPT's causal reasoning capabilities, involving four state-of-the-art (SOTA) versions of ChatGPT: **text-davinci-002**, **text-davinci-003**, **gpt-3.5-turbo** and **gpt-4**.

Firstly, we utilize the **Event Causality Identification (ECI)** task as a comprehensive causal reasoning benchmark. As shown in Figure 1, the ECI

task aims to determine whether there is a causal relationship between two events in a sentence. This requires the ChatGPT to not only use extensive commonsense knowledge but also understand the complex context composed of multiple entities and events. Finally, ChatGPT must combine all information to identify causal relationships.

Secondly, we employ **Causal Discovery (CD)** task for evaluation, which requires ChatGPT to possess a broader and more specialized knowledge, yet does not necessitate consideration of complex contexts. As shown in Figure 1, two CD task formats are used: 1) *multiple-choice*, aims to select the cause or effect of the input event from two options; 2) *binary classification*, aims to determine whether there is a causal relationship between the two input events. For *binary classification*, we convert each multiple-choice example into two binary-classification examples by pairing the input event with each of the two options. Our experiment indicates that *binary classification* is a more reliable evaluation method for ChatGPT.

Furthermore, as shown in Figure 1, we conduct **Causal Explanation Generation (CEG)** task to test whether ChatGPT can generate explanations for causal relations between events. This is typically used to test whether machines truly understand the principles behind causality, which is crucial for building a reliable causal reasoning system.

**Key takeaways**  The key findings and insights are summarized as follows:

- ChatGPT is not a good causal reasoner, but a good causal explainer.

- ChatGPT has a serious causal hallucination issue, where it tends to assume causal relationships between events, regardless of whether those relationships actually exist.

- The main reason of ChatGPT's causal hallucinations may be the reporting biases between causal and non-causal relationships in natural language, as well as ChatGPT's upgrading processes, such as RLHF (Ouyang et al., 2022b). Besides, the ICL and CoT (Wei et al., 2022) prompts can further exacerbate the causal hallucination of ChatGPT.

- The causal reasoning ability of ChatGPT is sensitive to the words used to express the causal concept in the prompt.

- As the number of events in the sentence increases, and the lexical distance between events becomes greater, ChatGPT's causal reasoning performance decreases. Besides, ChatGPT is better at identifying explicit causality rather than implicit causality.

- Open-ended generation prompts cannot improve ChatGPT's causal reasoning ability.

## 2  Related Work

### 2.1  Causal Reasoning in NLP

Causal reasoning ability is important in NLP. Previous work has made significant efforts to improve the causal reasoning ability of machines, such as incorporating external knowledge (Liu et al., 2020; Du et al., 2021; Liu et al., 2023; Wang et al., 2023), conducting causal-specific pre-training (Li et al., 2021; Zhou et al., 2022), or applying data augmentation techniques (Li et al., 2020; Zuo et al., 2021a,b). However, these methods are highly dependent on annotated training data in specific domains and task formats. They perform poorly in scenarios where annotated data is scarce. In this paper, we evaluate ChatGPT, which does not require training data.

### 2.2  Evaluation of ChatGPT's Capabilities

Recently, a large amount of work has conducted evaluations of ChatGPT's various capabilities. However, ChatGPT's causal reasoning capabilities have not been fully evaluated. Qin et al. (2023) and Chan et al. (2023) only employed the *multiple-choice* CD format on the COPA dataset to evaluate ChatGPT, which consists of only 1,000 examples primarily focused on simple everyday causality. Besides, our experiments show that this *multiple-choice* format leads to an overestimation of ChatGPT's performance. Furthermore, Kıcıman et al. (2023) claimed that ChatGPT achieved a 97% accuracy on the causal discovery task. However, they only required ChatGPT to determine the causal direction of 583 causal event pairs, without requiring it to predict whether causality exists. This does not constitute a complete causal discovery task.

In summary, previous evaluations only involved small-scale datasets and simple task formats, which overestimated ChatGPT's causal reasoning abilities. However, we conduct a comprehensive and objective evaluation of the ChatGPT's causal reasoning abilities, involving four different task forms and five widely-used causal reasoning datasets.

## 3 Evaluation Settings

### 3.1 Datasets and Evaluation Metrics

#### 3.1.1 Event Causality Identification

We conduct experiments on three widely used ECI datasets: 1) **EventStoryLine** v0.9 (ESC) (Caselli and Vossen, 2017), which contains 22 topics, 258 documents, 5,334 events and 1,770 causal event pairs; 2) **Causal-TimeBank** (CTB) (Mirza et al., 2014), which contains 184 documents, 6,813 events and 318 causal event pairs ; 3) **MAVEN-ERE** (Wang et al., 2022), which contains 90 topics, 4,480 documents, 103,193 events and 57,992 causal event pairs. Following previous works (Gao et al., 2019; Choubey and Huang, 2017), only the top 20 topics of ESC are used for evaluation. Besides, since MAVEN-ERE did not release the test set, we evaluate ChatGPT on its development set. We adopt Accuracy, Precision (P), Recall (R), and F1-score (F1) as the evaluation metrics.

#### 3.1.2 Causal Discovery

We conduct experiments on two widely used CD datasets: 1) **COPA** (Roemmele et al., 2011), which is a classic dataset for causal reasoning and consists of 1,000 multiple-choice questions that primarily focus on everyday life scenarios. 2) **e-CARE** (Du et al., 2022), contains 21,324 multiple-choice questions covering a wide range of domains. We adopt Accuracy as the evaluation metric.

#### 3.1.3 Causal Explanation Generation

We conduct experiments on **e-CARE**, which contains human annotated causal generations for 21,324 causal event pairs. Following the evaluation settings of e-CARE, we first adopt average-BLEU (n=4) (Papineni et al., 2002) and ROUGE-L (Lin, 2004) as the automatic evaluation metrics. Secondly, we sample 100 explanations generated by each version of ChatGPT on the **e-CARE** for human evaluation. Specifically, we label whether the generated explanation can explain the corresponding causal fact to calculate the accuracy.

### 3.2 Experiment Setting

For ChatGPT, we follow an instruction-prompt scheme for ECI, CD and CEG tasks. Figure 1 shows the prompts employed for these three causal reasoning tasks. We evaluate ChatGPT's performance under zero-shot settings. Additional prompts and settings are discussed in §5.

We conduct our experiments using OpenAI's official API[1], covering four progressive SOTA versions of ChatGPT: **text-davinci-002**, **text-davinci-003**, **gpt-3.5-turbo** and **gpt-4**. Specifically, **text-davinci-002** was further trained using RLHF to obtain **text-davinci-003**, which was subsequently further trained using conversational data to obtain **gpt-3.5-turbo**. Although OpenAI has not disclosed **gpt-4**, **gpt-4** has shown superior reasoning capabilities in various NLP tasks. For **gpt-4**, we sample 1000 instances from each dataset for evaluation. We set the temperature parameter to 0 to minimize randomness.

### 3.3 Baselines

**In this paper, all baseline methods for the three causal reasoning tasks are based on PLMs fine-tuned on the full training dataset.**

For the ECI and the CD task, we compare Chat-GPT with vanilla classification models based on **BERT-Base** (Devlin et al., 2019) and **RoBERTa-Base** (Liu et al., 2019). Their framework and training process are consistent with previous work (Liu et al., 2020; Li et al., 2021).

Besides, we compare ChatGPT with two SOTA ECI method: **KEPT** (Liu et al., 2023), based on BERT-Base, incorporated background and relational information for causal reasoning; and **DPJL** (Shen et al., 2022) based on RoBERTa-Base, introduced information about causal cue words and the relation between events into ECI model.

For the CEG task, we first compare ChatGPT with **GRU-based Seq2Seq model** (Chung et al., 2014) and **GPT2** (Radford et al., 2019). Their framework and training process are consistent with previous work (Du et al., 2022). Besides, we employ recent LLM baselines by finetuning **LLaMA 7B** (Touvron et al., 2023) and **FLAN-T5 11B** (Chung et al., 2022) on the e-CARE dataset.

## 4 Experimental Results

### 4.1 Event Causality Identification

Table 1 show the results on the three ECI datasets: ESC, CTB and MAVEN-ERE. We find that:

Firstly, ChatGPT, even the **gpt-4**, has been comprehensively outperformed by baseline methods based on fine-tuned small PLMs. This indicates that **ChatGPT is not a good causal reasoner in complex causal reasoning task like ECI.**

---

[1] https://platform.openai.com/ (accessed between 3/1/2023 and 4/28/2023)

| Methods | ESC | | | CTB | | | MAVEN-ERE | | |
|---|---|---|---|---|---|---|---|---|---|
| | **P** | **R** | **F1** | **P** | **R** | **F1** | **P** | **R** | **F1** |
| **BERT-Base (Devlin et al., 2019)** | 38.1 | 56.8 | 45.6 | 41.4 | 45.8 | 43.5 | 52.5 | 75.6 | 61.9 |
| **RoBERTa-Base (Liu et al., 2019)** | 42.1 | 64.0 | 50.8 | 39.9 | 60.9 | 48.2 | **52.8** | 75.1 | **62.0** |
| **KEPT (Liu et al., 2023)** | 50.0 | 68.8 | 57.9 | 48.2 | 60.0 | 53.5 | - | - | - |
| **DPJL (Shen et al., 2022)** | **65.3** | 70.8 | **67.9** | **63.6** | 66.7 | **64.6** | - | - | - |
| text-davinci-002 | 23.2 | 80.0 | 36.0 | 5.0 | 75.2 | 9.3 | 19.6 | **92.9** | 32.4 |
| text-davinci-003 | 33.2 | 74.4 | 45.9 | 8.5 | 64.4 | 15.0 | 25.0 | 75.1 | 37.5 |
| gpt-3.5-turbo | 27.6 | 80.2 | 41.0 | 6.9 | 82.6 | 12.8 | 19.9 | 85.8 | 32.3 |
| gpt-4 | 27.2 | **94.7** | 42.2 | 6.1 | **97.4** | 11.5 | 22.5 | 92.4 | 36.2 |
| | **Pos** | **Neg** | **Full** | **Pos** | **Neg** | **Full** | **Pos** | **Neg** | **Full** |
| **BERT-Base (Devlin et al., 2019)** | 59.5 | **83.6** | 79.7 | 46.4 | **87.8** | **86.2** | 75.7 | 86.9 | 85.1 |
| **RoBERTa-Base (Liu et al., 2019)** | 63.8 | 82.8 | **79.8** | 62.3 | 86.4 | 85.5 | 76.6 | **87.1** | **85.4** |
| text-davinci-002 | 80.0 | 43.1 | 49.6 | 75.2 | 41.9 | 43.2 | **92.9** | 21.2 | 33.5 |
| text-davinci-003 | 74.4 | 67.7 | 68.9 | 64.4 | 71.9 | 71.6 | 75.1 | 53.6 | 57.2 |
| gpt-3.5-turbo | 80.2 | 54.4 | 59.0 | 82.6 | 55.0 | 56.0 | 85.8 | 28.5 | 38.3 |
| gpt-4 | **94.7** | 41.4 | 51.4 | **97.4** | 39.1 | 41.4 | 92.4 | 33.9 | 44.0 |

Table 1: Experimental results (%) on the ECI task. P, R and F1 indicate Precision, Recall and F1-score, respectively. Pos, Neg and Full indicate accuracy on the causal pairs, non-causal pairs and all test datas, respectively.

| Methods | Multiple Choice | | Binary Classification | | | | | |
|---|---|---|---|---|---|---|---|---|
| | **e-CARE** | **COPA** | **e-CARE** | | | **COPA** | | |
| | **Full** | **Full** | **Pos** | **Neg** | **Full** | **Pos** | **Neg** | **Full** |
| **BERT-Base (Devlin et al., 2019)** | 75.4 | 75.4 | 65.7 | 95.2 | 65.0 | 59.5 | 88.9 | 54.5 |
| **RoBERTa-Base (Liu et al., 2019)** | 70.7 | 80.5 | 64.6 | 74.6 | 64.5 | 66.0 | 69.0 | 63.5 |
| text-davinci-002 | 78.4 | 94.4 | 18.5 | **95.2** | 56.8 | 55.6 | **92.4** | 74.0 |
| text-davinci-003 | 76.7 | 93.2 | 41.0 | 86.4 | 63.7 | 80.4 | 82.3 | **81.4** |
| gpt-3.5-turbo | 79.1 | 96.3 | 75.5 | 66.9 | 71.2 | 96.3 | 43.2 | 69.7 |
| gpt-4 | **84.5** | **98.1** | 84.8 | 57.5 | **71.2** | 97.9 | 38.5 | 68.2 |

Table 2: Experimental results (%) on the CD task. Pos, Neg and Full indicate accuracy on the causal pairs, non-causal pairs and all test datas, respectively.

Secondly, the recall of ChatGPT is high, but the precision is low, indicating that a large number of non-causal event pairs are falsely identified as causal pairs. This is also why ChatGPT performs particularly poorly on the CTB dataset, which contains more non-causal event pairs. The main reason for this may be that natural language contains a large number of descriptions of causal relationships, mainly indicated by causal cue words such as "lead to" and "therefore". However, natural language generally does not express which events are not causally related. Furthermore, since ChatGPT's ability comes from training on massive amounts of natural language text, such **reporting bias between causal and non-causal event pairs in texts makes ChatGPT good at identifying causal event pairs but not in recognizing non-causal event pairs.**

Besides, it can be observed that the fine-tuned small PLMs do better at identifying non-causal event pairs rather than causal ones. This is because there are much more negative examples than positive examples in the ECI dataset, and the fine-tuned models have learned such data distribution.

## 4.2 Causal Discovery

Table 2 show the results on the two CD datasets: COPA and e-CARE. We find that:

Firstly, although ChatGPT performs well in *multiple-choice*, its performance is poor in *binary classification*. The main reason is that in *multiple-choice*, ChatGPT only needs to consider the option that shows the more obvious causal or non-causal relationship with the input event, while the other more difficult option can be ignored. However, previous work (Qin et al., 2023; Chan et al., 2023) only used *multiple-choice* to evaluate ChatGPT's causal reasoning ability, **leading to a misrepresentation that ChatGPT is good at causal reasoning.**

Secondly, compared to the ECI task, ChatGPT achieves higher accuracy on non-causal pairs in the

| Methods | e-CARE | | |
|---|---|---|---|
| | AVG-BLEU | ROUGE-l | Human Evaluation |
| GRU-Seq2Seq (Chung et al., 2014) | 18.7 | 21.3 | 0.0 |
| GPT2 (Radford et al., 2019) | 32.0 | 31.5 | 20.0 |
| LLaMA 7B (Touvron et al., 2023) | **40.3** | 37.1 | 63.0 |
| FLAN-T5 11B (Chung et al., 2022) | 38.5 | **42.8** | 66.0 |
| text-davinci-003 | 10.55 | 37.49 | 83.0 |
| gpt-3.5-turbo | 7.32 | 40.31 | 82.0 |
| gpt-4 | 6.47 | 39.77 | 85.0 |
| Human Generation (Du et al., 2022) | 35.51 | 33.46 | **89.5** |

Table 3: Experimental results (%) on the CEG task. "Human Generation" is the human annotated explanations provided by Du et al. (2022).

CD task. This is mainly because the non-causal pairs in the e-CARE and COPA datasets are generated manually given a input event, and they have a simple structure and weak correlation with the input events, making them easier to identify. This is also the reason why the fine-tuned small PLMs do better at identifying non-causal event pairs rather than causal ones.

Besides, compared to the ECI task, ChatGPT achieves slightly lower accuracy on causal pairs in the e-CARE dataset. This is because e-CARE requires ChatGPT to grasp a wider range of knowledge, which involves not only commonsense knowledge in more scenarios, but also professional knowledge in certain fields, such as biology.

More importantly, we notice that the upgrading process of ChatGPT (**text-davinci-003→gpt-3.5-turbo→gpt-4**) leads to the ChatGPT become increasingly inclined to classify events as having a causal relationship, regardless of whether it is actually correct or not. This may be due to the alignment tax (Ouyang et al., 2022a) of RLHF. This indicates that **while OpenAI (2023) mentioned that ChatGPT's upgrading process reduces the hallucination issue in various other tasks, it also makes the ChatGPT better at fabricating causal relationships.** A preliminary analysis of the impact of RLHF on causal reasoning is provided in the Appendix A.

### 4.3 Causal Explanation Generation

Table 3 show the experimental results on the CEG task. It can be observed that:

Firstly, according to the human evaluation results, the accuracy of causal explanations generated by ChatGPT is close to those generated by humans. This indicates that **ChatGPT is a good causal explainer**.

Secondly, compared to "Human Generation", ChatGPT achieves a better ROUGE-l score, which is a text generation metric similar to "recall" in text classification. This is because **ChatGPT tends to generate explanations that are more complete and detailed.** This was confirmed by the evaluators during our human evaluation process. This is also the reason why ChatGPT received a lower AVG-BLEU score, which is a text generation metric similar to "precision" in text classification.

Thirdly, in manual evaluation, we find that the explanations generated by LLaMA and FLAN-T5 are highly correlated with the input events. However, the explanations might be mere repetitions of the input events or provide relevant but uninformative descriptions that cannot be used for explanation. This is also a reason for the poor performance of both LLaMA and FLAN-T5 in human evaluations.

Besides, compared to ChatGPT, the explanations provided by LLaMA and FLAN-T5 are noticeably shorter, as the gold explanations provided by e-CARE are very concise. However, ChatGPT excels in providing more comprehensive and detailed explanations in the zero-shot setting. This shows the advantage of ChatGPT's causal explanations compared to traditional fine-tuning methods.

Moreover, it is worth noting that fine-tuned LLaMA, FLAN-T5 and ChatGPT achieve similar ROUGE-l scores, but the two finetuned LLMs perform worse in our human evaluation. This is because that the fine-tuned LLaMA and FLAN-T5 may generate less informative explanations, e.g., mere repetitions of the input events. However, ChatGPT may offer valuable explanations for the input causal event pairs, but from different perspectives or in distinct syntactic forms compared to the gold explanations in e-CARE.

| Methods | ICL Settings | ESC | | |
|---|---|---|---|---|
| | | Pos | Neg | Full |
| text-davinci-002 | none | 80.0 | 43.1 | 49.6 |
| | 1 pos + 2 neg | 85.1 | 43.8 | 51.2 |
| | 2 pos + 2 neg | 88.4 | 37.3 | 46.4 |
| | 1 pos + 4 neg | 88.7 | 38.3 | 47.2 |
| | 2 pos + 4 neg | 89.7 | 36.0 | 45.5 |
| | 4 pos + 2 neg | 78.8 | 51.0 | 55.9 |
| gpt-3.5-turbo | none | 80.2 | 54.4 | 59.0 |
| | 1 pos + 2 neg | 94.4 | 26.6 | 38.7 |
| | 2 pos + 2 neg | 91.3 | 35.7 | 45.6 |
| | 1 pos + 4 neg | 92.6 | 31.0 | 41.9 |
| | 2 pos + 4 neg | 92.0 | 36.0 | 45.9 |
| | 4 pos + 2 neg | 82.2 | 50.3 | 56.0 |
| text-davinci-003 | none | 74.4 | 67.7 | 68.9 |
| | 1 pos + 2 neg | 92.4 | 35.3 | 45.2 |
| | 2 pos + 2 neg | 96.3 | 26.1 | 38.5 |
| | 1 pos + 4 neg | 95.9 | 27.1 | 39.3 |
| | 2 pos + 4 neg | 98.4 | 18.2 | 32.4 |
| | 4 pos + 2 neg | 92.8 | 38.2 | 47.9 |
| | 4 pos + 48 neg | 71.7 | 77.0 | 76.2 |
| | 40 pos + 20 neg | 86.1 | 59.1 | 63.4 |
| | top 1 similar | 85.3 | 50.1 | 55.6 |
| | top 4 similar | 95.6 | 28.0 | 38.8 |

Table 4: Performance of ChatGPT on the ECI task with ICL. "none" indicates ChatGPT without ICL.

| Methods | ICL Settings | e-CARE | | |
|---|---|---|---|---|
| | | Pos | Neg | Full |
| text-davinci-003 | none | 41.0 | 86.4 | 63.7 |
| | 1 pos + 2 neg | 90.3 | 41.3 | 65.8 |
| | 2 pos + 2 neg | 74.6 | 64.1 | 69.3 |
| | 1 pos + 4 neg | 83.4 | 55.5 | 69.5 |
| | 2 pos + 4 neg | 86.6 | 49.6 | 68.1 |
| | 4 pos + 2 neg | 72.7 | 66.5 | 69.6 |

Table 5: Performance of ChatGPT on the binary-classification CD task with ICL. "none" indicates ChatGPT without ICL.

# 5 Analysis

## 5.1 In-Context Learning

As shown in the Table 4 and Table 5, we analyze ChatGPT under different In-Context Learning settings: 1) "**x pos + y neg**": we randomly select $x$ positive and $y$ negative training examples as demonstrations for in-context learning, and all test cases share the same demonstrations; 2) "**top k similar**": for each test case, we retrieve the top $k$ most similar training examples as its demonstrations. To compute the similarity, we first encode them with BERT-large, then compute the cosine similarity of their embeddings. We further analyze the impact of the order and the label distribution of ICL demonstrations in Appendix B and C.

Firstly, when both $x$ and $y$ are less than or equal to 4, ICL mainly improves ChatGPT's accuracy on causal pairs, but decreases the accuracy on non-causal pairs. This may be because although ICL can stimulate the ChatGPT's abilities, ChatGPT is better at identifying causal event pairs. Therefore, ICL further exacerbates the imbalance of ChatGPT in identifying causal and non-causal pairs.

In addition, "4 pos + 48 neg" achieves higher Full accuracy. However, this is because it improves Neg accuracy at the expense of Pos accuracy, as the ESC dataset contains a larger proportion of non-causal pairs, making Neg accuracy have a greater impact on Full accuracy. A substantial improvement in the overall performance should not be a case of robbing Peter to pay Paul, rather than improve both the Pos and the Neg accuracy.

## 5.2 Chain-of-Thought Prompting

As shown in the Table 6, we analyze ChatGPT under different Chain-of-Thought settings: 1) "**-w/ CoT zero-shot**": we employ the zero-shot CoT (Kojima et al., 2022) by adding "Let's think step by step" after the prompt; 2) "**-w/ CoT x pos + y neg**": we manually annotate the reasoning chain for $x$ positive and $y$ negative training examples. They are selected as demonstrations for in-context learning, and all test cases share the same demonstrations. We further analyze the error types of ChatGPT in the Appendix D. The examples of our used demonstrations and the reasoning chains generated by ChatGPT are presented in the Appendix E.

Firstly, "-w/ CoT zero-shot" cannot effectively improve the performance of ChatGPT in the ECI task. This is because the quality of the reasoning chain generated by zero-shot CoT is not high enough to effectively guide the model.

Secondly, "-w/ CoT x pos + y neg" improves ChatGPT's accuracy on causal pairs, but decreases its accuracy on non-causal pairs. Observing the reasoning chains generated by ChatGPT, we found that the ChatGPT generates lower quality chains for non-causal pairs than for causal pairs. This difference may worsen the imbalance of ChatGPT in identifying causal and non-causal event pairs.

## 5.3 Ways of Expressing Causality in Prompts

As shown in Figure 2, we analyze the performance of ChatGPT on the ECI task using prompts that express the causal concept in different ways: 1) "**counterfactual**", a prompt based on the counterfactual causality view of Pearl (2009); 2) "**one-**

| Methods | ICL Settings | ESC | | | e-CARE | | |
|---|---|---|---|---|---|---|---|
| | | Pos | Neg | Full | Pos | Neg | Full |
| text-davinci-003 | none | 74.4 | 67.7 | 68.9 | 41.0 | 86.4 | 63.7 |
| -w/ CoT | zero-shot | 32.8 | 42.4 | 37.0 | 47.4 | 80.7 | 64.0 |
| | 4 pos + 2 neg | 97.1 | 41.1 | 50.1 | 84.6 | 51.1 | 67.8 |
| | 4 pos + 4 neg | 86.2 | 48.0 | 54.1 | 86.0 | 50.2 | 68.1 |
| | 4 pos + 8 neg | 76.8 | 62.4 | 64.7 | 82.9 | 52.9 | 67.9 |

Table 6: Performance of ChatGPT on the ECI and the binary-classification CD task with the Chain-of-Thought prompts. "none" indicates ChatGPT without ICL.

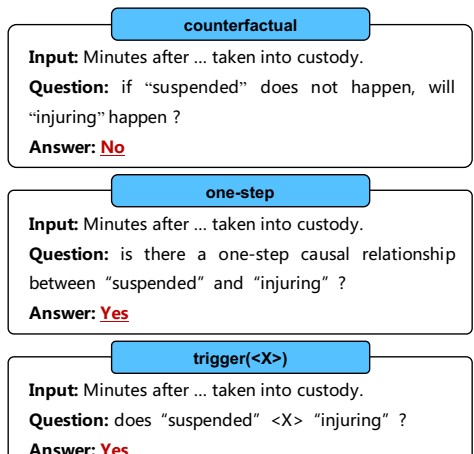

Figure 2: Prompts that express causal concepts in various ways. The content that requires ChatGPT to reply is marked in **red**.

| Methods | ESC | | |
|---|---|---|---|
| | Pos | Neg | Full |
| **text-davinci-003** | 74.4 | 67.7 | 68.9 |
| **-w/ counterfactual** | 98.5 | 00.8 | 16.3 |
| **-w/ one-step** | 44.4 | 91.7 | 84.4 |
| **-w/ trigger (lead to)** | 88.3 | 46.5 | 53.1 |
| **-w/ trigger (give rise to)** | 90.8 | 53.7 | 59.5 |
| **-w/ trigger (bring about)** | 87.1 | 53.1 | 58.4 |
| **-w/ trigger (result in)** | 83.3 | 59.5 | 63.3 |
| **-w/ trigger (in order to)** | 70.1 | 60.9 | 62.4 |
| **-w/ trigger (so that)** | 81.2 | 59.9 | 63.3 |

Table 7: Performance of ChatGPT in the ECI task using prompts that express the causality in different ways.

step", we add constraint words "one-step" to alleviate the issue of identifying non-causal event pairs as causal; and 3) "**trigger(<X>)**", we use different causal cue words <X> (e.g., "lead to") to construct prompts. Results are shown in Table 7.

Firstly, the "**counterfactual**" prompt makes almost all non-causal pairs to be identified as causal. This is mainly because ChatGPT's counterfactual reasoning results are not accurate.

Secondly, the "**one-step**" improves ChatGPT's accuracy on non-causal pairs, but lowers its accuracy on causal pairs. This is because while constraint words such as "one-step" can make the model more likely to predict event pairs as non-causal, it does not truly enhance ChatGPT's causal reasoning abilities.

Moreover, the performance of "**trigger(<X>)**" with different causal cue words is significantly different. This may be due to the fact that during pre-training, ChatGPT mainly learns causal knowledge triggered by causal cue words, but the distributions of causality triggered by each cue word are quite different. Therefore, causal cue words that have the same meaning to humans may represent different causal concepts to ChatGPT. This further indicates that it is challenging to accurately convey what causality means to ChatGPT through prompts.

### 5.4 Lexical Distance between Events

As shown in the Figure 3, we analyze the performance of ChatGPT on pairs of events with different lexical distances in the ECI task. The "lexical distance" refers to the number of words that separate two events within a sentence.

Firstly, we find that as the event distances increase, ChatGPT is more inclined to predict event pairs as non-causal. This may be because in natural language, the larger the distance between events, the less likely there is a causal relationship, and ChatGPT has learned this pattern.

Secondly, as the event distances increase, the F1 scores of ChatGPT decrease. This indicates that ChatGPT is not good at identifying long-distance causal relationships. An outlier is the F1 score of **gpt-4** at the interval [25,30). This is due to the fact that out of 1000 test instances for **gpt-4**, there are only 35 examples within the interval [25,30),

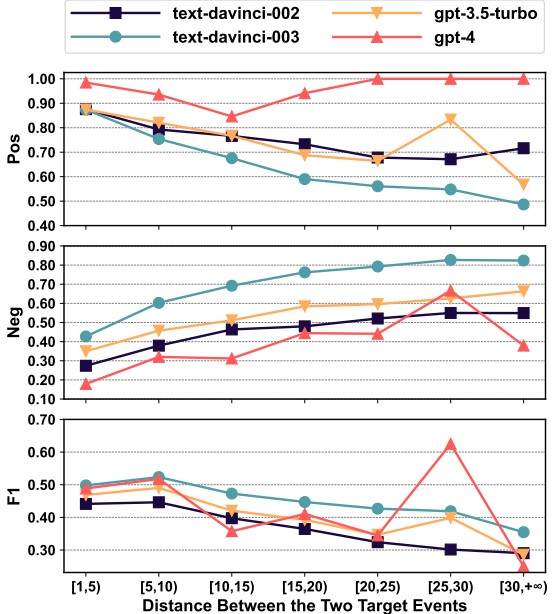

Figure 3: Performance of ChatGPT on pairs of events with different lexical distances in the ESC dataset.

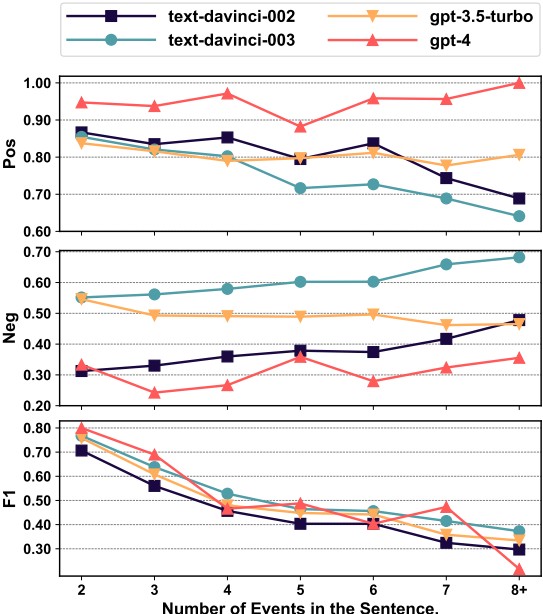

Figure 4: Performance of ChatGPT on sentences with different numbers of events in the ESC dataset.

leading to more random performance. However, all other results demonstrate that ChatGPTs' performance decreases as the event distance increases.

### 5.5 Density of Events

As shown in the Figure 4, we analyze the performance of ChatGPT in sentences with different numbers of events in the ECI task. We find that:

Firstly, as the event density increases, most versions of ChatGPT are more inclined to predict event pairs as non-causal. This is mainly because as the event density increases, the context of events becomes more complex, making it more difficult to capture the correlations between events.

Secondly, as the event density increases, the F1 scores of ChatGPT decreases. This indicates that ChatGPT is not good at handling complex situations involving multiple events.

### 5.6 Types of Causal Relationship

As shown in the Figure 5, we analyze the accuracy of ChatGPT on pairs of events with different types of causal relationships in the ECI task: 1) **Explicit Causality**, which refers to causal relationships explicitly triggered by causal cue words (e.g., "lead to"); 2) **Implicit Causality**, which refers to causal relationships expressed without causal cue words.

It can be observed that, compared to implicit causality, ChatGPT performs better on capturing explicit causality. This is mainly because iden-

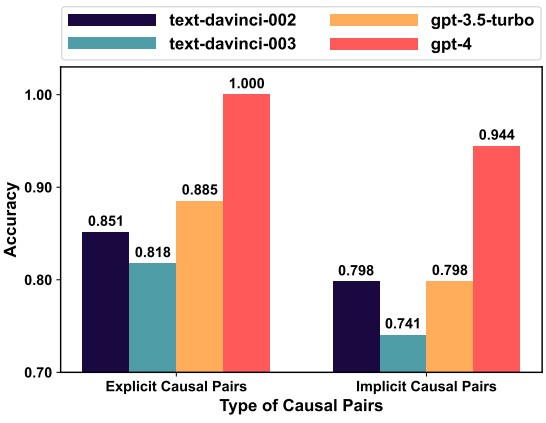

Figure 5: Performance of ChatGPT on pairs of events with different types of causality in the ESC dataset.

tifying explicit causality only requires recognizing causal cue words, whereas identifying implicit causality requires reasoning with contextual information and commonsense knowledge.

### 5.7 Prompts in the Form of Open-Ended Generation

Recently, Arora et al. (2023) revealed that open-ended prompts ("Who went to the park?") tend to yield better results for ChatGPT than prompts that restrict ChatGPT's outputs ("John went to the park. True or False?"). As shown in Table 8, we analyze ChatGPT with open-ended prompts: 1) "**open-ended A.1/2/3**", requires ChatGPT to generate all the causal event pairs in the input sentence.

Figure 6: Prompts in the open-ended form. The content that requires ChatGPT to reply is marked in **red**.

| Methods | Prompt | ESC | | |
| --- | --- | --- | --- | --- |
| | | **P** | **R** | **F1** |
| | close-ended | 27.6 | 80.2 | 41.0 |
| **gpt-** | open-ended A.1 | 4.8 | 16.3 | 7.4 |
| **3.5-** | open-ended A.2 | 6.9 | 8.0 | 7.4 |
| **turbo** | open-ended A.3 | 14.1 | 10.4 | 12.0 |
| | open-ended B | 6.3 | 54.0 | 11.3 |

Table 8: Performance of ChatGPT on the ECI task with open-ended generation prompts. "close-ended" indicates the origin ECI prompt shown in Figure 1. It is worth noting that the "close-ended" prompt does not explicitly require ChatGPT to output Yes or No, but its question format guides ChatGPT almost always output Yes or No.

We designed three different prompts to fully evaluate ChatGPT's performance. 2) "**open-ended B**", gives a target event in the input sentence, and requires ChatGPT to generate events in the input sentence that have causal relations with the target event. We employ a relaxed P, R, and F1 calculation for open-ended prompts. Specifically, a predicted causal-effect pair is considered correct if at least one token is shared between the predicted and the labeled cause, as well as between the predicted and the labeled effect. The formats of these prompts are shown in Figure 6.

It can be observed that the open-ended prompts decrease the performance of ChatGPT. This is because open-ended prompts require ChatGPT to jointly perform the event extraction and the ECI task. However, previous studies (Gao et al., 2023; Wei et al., 2023) show that ChatGPT is not good at extracting events.

## 6 Conclusion

In this paper, we conduct a comprehensive evaluation of ChatGPT's causal reasoning capabilities. Experiments show that: 1) ChatGPT is not a good causal reasoner, but is good at causal explanation generation; 2) ChatGPT has a serious causal hallucination, possibly due to the causal reporting biases and ChatGPT's upgrading processes 3) The ICL and CoT techniques can further exacerbate such causal hallucination; 4) The ChatGPT is sensitive to the words used to express the causal concept in prompts, and open-ended causal reasoning prompts is not suitable for ChatGPT; 5) For events in sentences, ChatGPT excels at capturing explicit causality, and performs better in sentences with lower event density and smaller event distances.

Although there may be more delicate prompts that can further surpass our reported results, we believe that relying solely on prompts cannot fundamentally solve the issues that ChatGPT faces in causal reasoning. We hope that this study can inspire future works, such as addressing the causal hallucination issue of ChatGPT or further evaluating ChatGPT in scenarios involving multi-factor and multi-modal causal reasoning.

## Acknowledgements

We would like to thank Jianbai Zhao for his effort on prompt design and human evaluation, and the anonymous reviewers for their constructive comments, and gratefully acknowledge the support of the National Natural Science Foundation of China (U22B2059, 62176079), and the Natu-

ral Science Foundation of Heilongjiang Province
(YQ2022F005).

## Limitations

This work is a comprehensive evaluation on the
causal reasoning ability of ChatGPT, and it has
several limitations. Firstly, ChatGPT's capabilities
are constantly being updated, and current test re-
sults may change as ChatGPT evolves. Secondly,
although OpenAI has provided rough introductions
of different versions of ChatGPT, the implementa-
tion details are unclear, making it difficult to deeply
analyze why different versions of ChatGPT have
different performance, and how each data and train-
ing technique affects the performance of ChatGPT.
Finally, there may still be prompts that can further
outperform the results we reported, such as differ-
ent questioning formats and more advanced ICL
techniques, but we believe that relying solely on
prompts cannot fundamentally solve the illusion
problem that ChatGPT currently faces in causal
reasoning.

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

resentations*.

Yuntao Bai, Andy Jones, Kamal Ndousse, Amanda
Askell, Anna Chen, Nova DasSarma, Dawn Drain,
Stanislav Fort, Deep Ganguli, Tom Henighan,
Nicholas Joseph, Saurav Kadavath, Jackson Kernion,
Tom Conerly, Sheer El-Showk, Nelson Elhage, Zac
Hatfield-Dodds, Danny Hernandez, Tristan Hume,
Scott Johnston, Shauna Kravec, Liane Lovitt, Neel
Nanda, Catherine Olsson, Dario Amodei, Tom
Brown, Jack Clark, Sam McCandlish, Chris Olah,
Ben Mann, and Jared Kaplan. 2022. Training a help-
ful and harmless assistant with reinforcement learn-
ing from human feedback.

Tommaso Caselli and Piek Vossen. 2017. The event
StoryLine corpus: A new benchmark for causal and
temporal relation extraction. In *Proceedings of the
Events and Stories in the News Workshop*.

Chunkit Chan, Jiayang Cheng, Weiqi Wang, Yuxin
Jiang, Tianqing Fang, Xin Liu, and Yangqiu Song.
2023. Chatgpt evaluation on sentence level relations:
A focus on temporal, causal, and discourse relations.

Prafulla Kumar Choubey and Ruihong Huang. 2017. A
sequential model for classifying temporal relations
between intra-sentence events. In *Proc. of EMNLP*.

Hyung Won Chung, Le Hou, Shayne Longpre, Barret
Zoph, Yi Tay, William Fedus, Yunxuan Li, Xuezhi
Wang, Mostafa Dehghani, Siddhartha Brahma, Al-
bert Webson, Shixiang Shane Gu, Zhuyun Dai,
Mirac Suzgun, Xinyun Chen, Aakanksha Chowdh-
ery, Alex Castro-Ros, Marie Pellat, Kevin Robinson,
Dasha Valter, Sharan Narang, Gaurav Mishra, Adams
Yu, Vincent Zhao, Yanping Huang, Andrew Dai,
Hongkun Yu, Slav Petrov, Ed H. Chi, Jeff Dean, Ja-
cob Devlin, Adam Roberts, Denny Zhou, Quoc V. Le,
and Jason Wei. 2022. Scaling instruction-finetuned
language models.

Junyoung Chung, Caglar Gulcehre, KyungHyun Cho,
and Yoshua Bengio. 2014. Empirical evaluation of
gated recurrent neural networks on sequence model-
ing. *arXiv preprint arXiv:1412.3555*.

Jacob Devlin, Ming-Wei Chang, Kenton Lee, and
Kristina Toutanova. 2019. BERT: Pre-training of
deep bidirectional transformers for language under-
standing. In *Proc. of NAACL*.

Li Du, Xiao Ding, Kai Xiong, Ting Liu, and Bing Qin.
2021. ExCAR: Event graph knowledge enhanced
explainable causal reasoning. In *Proceedings of the
59th Annual Meeting of the Association for Compu-
tational Linguistics and the 11th International Joint
Conference on Natural Language Processing (Vol-
ume 1: Long Papers)*, pages 2354–2363, Online. As-
sociation for Computational Linguistics.

Li Du, Xiao Ding, Kai Xiong, Ting Liu, and Bing Qin.
2022. e-care: a new dataset for exploring explainable
causal reasoning. In *Proceedings of the 60th Annual
Meeting of the Association for Computational Lin-
guistics (Volume 1: Long Papers)*, pages 432–446.

Jun Gao, Huan Zhao, Changlong Yu, and Ruifeng Xu.
2023. Exploring the feasibility of chatgpt for event
extraction. *arXiv preprint arXiv:2303.03836*.

Lei Gao, Prafulla Kumar Choubey, and Ruihong Huang.
2019. Modeling document-level causal structures
for event causal relation identification. In *Proc. of
NAACL*.

Takeshi Kojima, Shixiang Shane Gu, Machel Reid, Yu-
taka Matsuo, and Yusuke Iwasawa. 2022. Large lan-
guage models are zero-shot reasoners. *arXiv preprint
arXiv:2205.11916*.

Emre Kıcıman, Robert Ness, Amit Sharma, and Chen-
hao Tan. 2023. Causal reasoning and large language
models: Opening a new frontier for causality.

Zhongyang Li, Xiao Ding, Kuo Liao, Bing Qin, and
Ting Liu. 2021. Causalbert: Injecting causal knowl-
edge into pre-trained models with minimal supervi-
sion. *arXiv preprint arXiv:2107.09852*.

Zhongyang Li, Xiao Ding, Ting Liu, J. Edward Hu,
and Benjamin Van Durme. 2020. Guided generation
of cause and effect. In *Proceedings of the Twenty-
Ninth International Joint Conference on Artificial*

*Intelligence, IJCAI-20*, pages 3629–3636. International Joint Conferences on Artificial Intelligence Organization. Main track.

Chin-Yew Lin. 2004. Rouge: A package for automatic evaluation of summaries. In *Text summarization branches out*, pages 74–81.

Jian Liu, Yubo Chen, and Jun Zhao. 2020. Knowledge enhanced event causality identification with mention masking generalizations. In *Proc. of IJCAI*.

Jintao Liu, Zequn Zhang, Zhi Guo, Li Jin, Xiaoyu Li, Kaiwen Wei, and Xian Sun. 2023. Kept: Knowledge enhanced prompt tuning for event causality identification. *Knowledge-Based Systems*, 259:110064.

Yinhan Liu, Myle Ott, Naman Goyal, Jingfei Du, Mandar Joshi, Danqi Chen, Omer Levy, Mike Lewis, Luke Zettlemoyer, and Veselin Stoyanov. 2019. Roberta: A robustly optimized bert pretraining approach. *arXiv preprint arXiv:1907.11692*.

Paramita Mirza, Rachele Sprugnoli, Sara Tonelli, and Manuela Speranza. 2014. Annotating causality in the TempEval-3 corpus. In *Proc. of EACL*.

OpenAI. 2023. Gpt-4 technical report.

Long Ouyang, Jeffrey Wu, Xu Jiang, Diogo Almeida, Carroll Wainwright, Pamela Mishkin, Chong Zhang, Sandhini Agarwal, Katarina Slama, Alex Ray, John Schulman, Jacob Hilton, Fraser Kelton, Luke Miller, Maddie Simens, Amanda Askell, Peter Welinder, Paul F Christiano, Jan Leike, and Ryan Lowe. 2022a. Training language models to follow instructions with human feedback. In *Advances in Neural Information Processing Systems*, volume 35, pages 27730–27744. Curran Associates, Inc.

Long Ouyang, Jeffrey Wu, Xu Jiang, Diogo Almeida, Carroll Wainwright, Pamela Mishkin, Chong Zhang, Sandhini Agarwal, Katarina Slama, Alex Ray, et al. 2022b. Training language models to follow instructions with human feedback. *Advances in Neural Information Processing Systems*, 35:27730–27744.

Kishore Papineni, Salim Roukos, Todd Ward, and Wei-Jing Zhu. 2002. Bleu: a method for automatic evaluation of machine translation. In *Proceedings of the 40th annual meeting of the Association for Computational Linguistics*, pages 311–318.

Judea Pearl. 2009. *Causality*. Cambridge university press.

Chengwei Qin, Aston Zhang, Zhuosheng Zhang, Jiaao Chen, Michihiro Yasunaga, and Diyi Yang. 2023. Is chatgpt a general-purpose natural language processing task solver?

Alec Radford, Jeff Wu, Rewon Child, David Luan, Dario Amodei, and Ilya Sutskever. 2019. Language models are unsupervised multitask learners. *arXiv preprint arXiv:1412.3555*.

Melissa Roemmele, Cosmin Adrian Bejan, and Andrew S Gordon. 2011. Choice of plausible alternatives: An evaluation of commonsense causal reasoning. In *AAAI spring symposium: logical formalizations of commonsense reasoning*, pages 90–95.

Shirong Shen, Heng Zhou, Tongtong Wu, and Guilin Qi. 2022. Event causality identification via derivative prompt joint learning. In *Proceedings of the 29th International Conference on Computational Linguistics*, pages 2288–2299, Gyeongju, Republic of Korea. International Committee on Computational Linguistics.

Hugo Touvron, Thibaut Lavril, Gautier Izacard, Xavier Martinet, Marie-Anne Lachaux, Timothée Lacroix, Baptiste Rozière, Naman Goyal, Eric Hambro, Faisal Azhar, Aurelien Rodriguez, Armand Joulin, Edouard Grave, and Guillaume Lample. 2023. Llama: Open and efficient foundation language models.

Xiaozhi Wang, Yulin Chen, Ning Ding, Hao Peng, Zimu Wang, Yankai Lin, Xu Han, Lei Hou, Juanzi Li, Zhiyuan Liu, Peng Li, and Jie Zhou. 2022. Maven-ere: A unified large-scale dataset for event coreference, temporal, causal, and subevent relation extraction. In *Proceedings of EMNLP*.

Zhaowei Wang, Quyet V. Do, Hongming Zhang, Jiayao Zhang, Weiqi Wang, Tianqing Fang, Yangqiu Song, Ginny Y. Wong, and Simon See. 2023. Cola: Contextualized commonsense causal reasoning from the causal inference perspective.

Jason Wei, Xuezhi Wang, Dale Schuurmans, Maarten Bosma, brian ichter, Fei Xia, Ed H. Chi, Quoc V Le, and Denny Zhou. 2022. Chain of thought prompting elicits reasoning in large language models. In *Advances in Neural Information Processing Systems*.

Xiang Wei, Xingyu Cui, Ning Cheng, Xiaobin Wang, Xin Zhang, Shen Huang, Pengjun Xie, Jinan Xu, Yufeng Chen, Meishan Zhang, Yong Jiang, and Wenjuan Han. 2023. Zero-shot information extraction via chatting with chatgpt.

Yucheng Zhou, Tao Shen, Xiubo Geng, Guodong Long, and Daxin Jiang. 2022. ClarET: Pre-training a correlation-aware context-to-event transformer for event-centric generation and classification. In *Proceedings of the 60th Annual Meeting of the Association for Computational Linguistics (Volume 1: Long Papers)*, pages 2559–2575, Dublin, Ireland. Association for Computational Linguistics.

Xinyu Zuo, Pengfei Cao, Yubo Chen, Kang Liu, Jun Zhao, Weihua Peng, and Yuguang Chen. 2021a. Improving event causality identification via self-supervised representation learning on external causal statement. In *Findings of the Association for Computational Linguistics: ACL-IJCNLP 2021*, pages 2162–2172, Online. Association for Computational Linguistics.

Xinyu Zuo, Pengfei Cao, Yubo Chen, Kang Liu, Jun Zhao, Weihua Peng, and Yuguang Chen. 2021b. LearnDA: Learnable knowledge-guided data augmentation for event causality identification. In *Proceedings of the 59th Annual Meeting of the Association for Computational Linguistics and the 11th International Joint Conference on Natural Language Processing (Volume 1: Long Papers)*, pages 3558–3571, Online. Association for Computational Linguistics.

| Methods | ESC | | | CTB | | | MAVEN-ERE | | |
|---|---|---|---|---|---|---|---|---|---|
| | **P** | **R** | **F1** | **P** | **R** | **F1** | **P** | **R** | **F1** |
| **FLAN-T5-Large ([Chung et al., 2022](#))** | 19.4 | 91.1 | 32.0 | 4.2 | 82.6 | 7.9 | 18.1 | 93.5 | 30.3 |
| - w/ RLHF | 17.6 | 97.9 | 29.8 | 3.9 | 99.0 | 7.5 | 17.1 | 99.4 | 29.2 |
| | **Pos** | **Neg** | **Full** | **Pos** | **Neg** | **Full** | **Pos** | **Neg** | **Full** |
| **FLAN-T5-Large ([Chung et al., 2022](#))** | 91.1 | 18.6 | 31.4 | 82.6 | 23.0 | 25.3 | 93.5 | 12.8 | 26.6 |
| - w/ RLHF | 97.9 | 1.5 | 18.6 | 99.0 | 0.9 | 4.7 | 99.4 | 0.8 | 17.6 |

Table 9: Experimental results (%) on the ECI task. P, R and F1 indicate Precision, Recall and F1-score, respectively. Pos, Neg and Full indicate accuracy on the causal pairs, non-causal pairs and all test datas, respectively.

## A  Effect of RLHF on Causal Reasoning

It is necessary to explore why RLHF enhances the causal hallucination issue of ChatGPT. Due to OpenAI's decision not to open-source ChatGPT, we lack access to specific details about the data and experimental setup for ChatGPT's RLHF. As an alternative approach, we analyze the effect of RLHF with the Anthropic RLHF dataset ([Bai et al., 2022](#)), which is an open-source RLHF dataset. As shown in Table 9, we test the zero-shot performance of FLAN-T5-Large ([Chung et al., 2022](#)) with/without RLHF process on the Anthropic dataset.

It can be find that the RLHF also exacerbates the causal hallucination issue of FLAN-T5-Large. Through our analysis of the Anthropic dataset, this may be due to:

1. Among questions about "Why", only 10.29% include the word "not". This indicates that the majority of RLHF data guide the LLMs about causality rather than non-causality.

2. In the gold responses, the frequency of the word "yes" is approximately twice that of the word "no". This could potentially increase the likelihood of the LLMs producing the positive label rather than the negative label.

Both of these two characteristics in the RLHF data could potentially exacerbate ChatGPT's causal hallucination, leading it to assume causal relationships between events, regardless of whether those relationships actually exist.

## B  Effect of the Label Distribution in ICL Demonstrations

As shown in Table 10, we analyze the impact of the labels of ICL demonstrations on performance of "top k similar" (described in §5.1). For k=4, we first divide the ESC dataset into five subsets,

| Demonstrations | Pos | Neg | Full | Proportion+ |
|---|---|---|---|---|
| **0 pos 4 neg** | 98.4 | 21.8 | 26.7 | 6.4 |
| **1 pos 3 neg** | 88.8 | 27.5 | 39.1 | 18.9 |
| **2 pos 2 neg** | 88.6 | 36.3 | 49.5 | 25.2 |
| **3 pos 1 neg** | 89.5 | 47.1 | 66.9 | 46.6 |
| **4 pos 0 neg** | 95.3 | 35.7 | 83.7 | 80.6 |

Table 10: Experimental results (%) of "top 4 similar" in §5.1 with different labels of demonstrations on the ESC dataset.

each containing instances that only use 0, 1, 2, 3, or 4 causal demonstrations in their top 4 similar demonstrations, respectively. Then, we present the performance of the "top 4 similar" on these five subsets. "Proportion+" is the proportion of causal instances in the corresponding subset. "x pos y neg" indicates the subset that only use x causal and y non-causal demonstrations.

Firstly, we can observe that when using only causal or non-causal demonstrations, the model achieves a higher Pos accuracy. This might be because that including only one classes prevents the model from contrasting the meanings of different labels, thus potentially confusing the model's understanding of the task objectives.

Additionally, when using both causal and non-causal demonstrations, there is a smaller change in Pos accuracy, while Neg accuracy increases as the number of causal demonstrations rises. This might be because having more causal demonstrations helps the model understand the situations that truly involve causality, thus avoiding misclassifying non-causal instances as causal.

## C  Effect of the Order of ICL Demonstrations

As shown in Table 12, we analyze the few-shot ChatGPT's performance under different orders of ICL demonstrations: 1) **Causal first**: from causal demonstrations to non-causal demonstrations; 2)

| Error Type | text-davinci-003 | gpt-3.5-turbo | gpt-4 |
|---|---|---|---|
| **Fake conditions** | 31 | 39 | 45 |
| **Incorrect basic event relationships** | 19 | 17 | 16 |
| **Wrong target event localization** | 15 | 11 | 13 |
| **Incorrect commonsense knowledge** | 12 | 15 | 9 |
| **Other** | 23 | 18 | 17 |

Table 11: The distribution (%) of causal reasoning error types of ChatGPT.

| Demonstrations | Pos | Neg | Full |
|---|---|---|---|
| **Zero-shot** | 74.4 | 67.7 | 68.9 |
| **Causal first** | 97.1 | 23.4 | 35.1 |
| **Non-causal first** | 92.8 | 29.0 | 39.2 |
| **Random1** | 95.6 | 28.0 | 38.8 |
| **Random2** | 95.7 | 27.8 | 38.6 |
| **Random3** | 95.7 | 27.4 | 38.3 |

Table 12: Experimental results (%) of "top 4 similar" in §5.1 with different orders of demonstrations on the ESC dataset.

**Non-causal first**: from non-causal demonstrations to causal demonstrations; 3) **Random**: we conduct three times experiments under random demonstration orders. Besides, **Zero-shot** indicates the performance of ChatGPT under the zero-shot setting.

Firstly, we find that **Causal first** is more inclined to classify event pairs as causal compared to **Non-causal first**. This might be because that the demonstrations located earlier have a stronger impact on the ChatGPT.

Secondly, despite different orders, all of these few-shot settings make the ChatGPT more inclined to classify event pairs as causal compared to the zero-shot setting.

## D    Error Analysis for the Causal Reasoning of ChatGPT

As shown in Table 11, we analyze the error types of ChatGPT's causal reasoning by observing the reasoning chains generated in the CoT setting. Specifically, we randomly select 100 instances of errors for each model on the ESC dataset, and then manually annotate them to analyze the types of errors among different models. Common error types include: 1) Fabricating additional fake conditions to establish causality, even if these conditions are not described or are incorrect in the input sentence; 2) Misunderstanding basic event relationships such as sub-events and temporal relationships between events; 3) Failing to accurately identify which two events the causal question is referring to; 4) Introducing incorrect commonsense knowledge.

Firstly, causal reasoning is a comprehensive skill that requires commonsense knowledge, as well as the ability to understand basic event relationships, and to perform logical reasoning based on information. However, ChatGPT is still not entirely reliable in these aspects, leading to the accumulation of errors. On the other hand, ChatGPT has encountered numerous causal event pairs in pre-training data, enabling it to associate many event pairs with potentially causal contexts. However, this context might not align with the input.

Besides, compared to **text-davinci-003**, which is not fine-tuned on dialog data, **gpt-3.5-turbo** and **gpt-4** show a clearer tendency to fabricate additional fake conditions to establish causality. This could be due to dialog data guiding them to produce longer and more divergent responses, which deviate from the context provided in the original input. Additionally, **gpt-4** introduces fewer incorrect commonsense knowledge, as it has a better grasp of knowledge.

## E    Details of CoT Experiments

Figure 7 shows the examples of the demonstrations utilized in our few-shot CoT experiments. Figure 8 shows the examples of the reasoning chains generated by ChatGPT.

## COT Demonstrations for the ECI Task

**INPUT:** An earthquake with a magnitude of at least 5 . 9 hit southern Iran on Sunday , killing at least 10 people and injuring many others .
**Question:** In the INPUT sentence, is there a causal relationship between "injuring" and "earthquake"?
**Reasoning Process:**
1. From the INPUT, An earthquake hit southern Iran
2. From the INPUT, the earthquake is with a magnitude of at least 5 . 9
3. a magnitude of at least 5 . 9 is dangerous enough for destroying buildings and causing things to fall.
4. Typically, high-magnitude earthquakes cause casualties.
5. From the INPUT, there have been people who is injured.
6. Therefore, there is a very high probability that the injuring is caused by the earthquake in the input sentence.
**Answer:** Yes.

**INPUT:** Power was restored to the afflicted villages on the Gulf island of Qeshm after a blackout caused by the quake , which struck on Sunday with a force of about 6 . 0 on the Richter scale .
**Question:** In the INPUT sentence, is there a causal relationship between "struck" and "quake"?
**Reasoning Process:**
1. The quake is the subject of verb struck, and they together form the event that the earthquake struck the villages.
2. Therefore, they are different component of one event.
3. causal relationship is between two events, rather than a verb and its subject.
4. Therefore, there is not a causal relationship between "struck" and "quake".
**Answer:** No.

## COT Demonstrations for the CR Task

**Event A:** This phenomenon is called speciation.
**Event B:** A species gradually adapted to its environment.
**Question:** is there a causal relationship between Event A and Event B ?
**Reasoning Process:**
1. Based on the expertise in the field of biology, speciation is often the result of environmental adaptation.
2. Therefore, there is causal relationship between Event A and Event B.
**Answer:** Yes

**Event A:** The leader of the orphanage decided to find a kind person for her to adopt her.
**Event B:** Her mother abandoned her.
**Question:** is there a causal relationship between Event A and Event B ?
**Reasoning Process:**
1. There is no correlation between Event A and Event B.
2. If there is no correlation, then there must be no causal relationship.
3. Therefore, there is no causal relationship between Event A and Event B.
**Answer:** No

Figure 7: Examples of the CoT demonstrations.

## Reasoning Chains Generated by ChatGPT for the ECI Task

**1. Correct Answer: Yes**
**INPUT:** Tens of thousands of people took to the streets in a nationwide strike to protest new taxes and government spending cuts demanded by the International Monetary Fund and other European nations before heavily indebted Greece gets a euro110 billion ( $141 billion ) loan package to keep it from defaulting .
**Question:** In the INPUT sentence, is there a causal relationship between "protest" and "keep"?
**Reasoning Process:**
1. From the INPUT, tens of thousands of people took to the streets in a nationwide strike to protest new taxes and government spending cuts.
2. From the INPUT, the European nations demanded these austerity measures in order to "keep" Greece from defaulting.
3. Typically, people protest or strike as a form of resistance against certain event.
4. From the INPUT, the event is the austerity measures demanded by the European nations.
5. Therefore, there is a causal relationship between "protest" and "keep".
**Answer:** Yes.

**2. Correct Answer: No**
**INPUT:** A man who was fired last week from his job at an accounting firm in suburban Detroit walked into his former office with a shotgun on Monday and shot three people , killing one , before fleeing and leading police officers on a high - speed chase , the authorities said .
**Question:** In the INPUT sentence, is there a causal relationship between "fleeing" and "walked"?
**Reasoning Process:**
1. From the INPUT, the man walked into his former office with a shotgun.
2. From the INPUT, the man fled and lead police officers on a high-speed chase.
3. Typically, when people are facing with an intimidating situation, they will fled.
4. From the INPUT, the intimidating situation is walking into his former office with a shotgun.
5. Therefore, there is a causal relationship between "fleeing" and "walked".
**Answer:** Yes.

## Reasoning Chains Generated by ChatGPT for the CR Task

**1. Correct Answer: Yes**
**Event A:** The swimmer saw shark fins in this water.
**Event B:** He chose to leave here.
**Question:** is there a causal relationship between Event A and Event B ?
**Reasoning Process:**
1. Seeing shark fins in the water is a sign of danger and can cause fear in a swimmer.
2. Therefore, there is causal relationship between Event A and Event B.
**Answer:** Yes

**2. Correct Answer: No**
**Event A:** The teacher wants the students to think spontaneously.
**Event B:** Various elimination processes was adopted to achieve that.
**Question:** is there a causal relationship between Event A and Event B ?
**Reasoning Process:**
1. The elimination processes are used to encourage students to think spontaneously.
2. Therefore, there is causal relationship between Event A and Event B.
**Answer:** Yes

Figure 8: Examples of the reasoning chains generated by ChatGPT.