# OpenReview forum: "Is ChatGPT a Good Causal Reasoner? A Comprehensive Evaluation"
_EMNLP/2023/Conference — EMNLP 2023 Findings_

### Official Review · Reviewer_vWs8 · 2023-07-28

**Soundness:** 3

**Excitement:**

3: Ambivalent: It has merits (e.g., it reports state-of-the-art results, the idea is nice), but there are key weaknesses (e.g., it describes incremental work), and it can significantly benefit from another round of revision. However, I won't object to accepting it if my co-reviewers champion it.

**Missing References:**

[1] Bang, Y., Cahyawijaya, S., Lee, N., Dai, W., Su, D., Wilie, B., ... & Fung, P. (2023). A multitask, multilingual, multimodal evaluation of chatgpt on reasoning, hallucination, and interactivity. arXiv preprint arXiv:2302.04023.

[2] Manakul, P., Liusie, A., & Gales, M. J. (2023). Selfcheckgpt: Zero-resource black-box hallucination detection for generative large language models. arXiv preprint arXiv:2303.08896.

[3] Zhao, Z., Wallace, E., Feng, S., Klein, D., & Singh, S. (2021, July). Calibrate before use: Improving few-shot performance of language models. In International Conference on Machine Learning (pp. 12697-12706). PMLR.

[4] Yao Lu, Max Bartolo, Alastair Moore, Sebastian Riedel, and Pontus Stenetorp. 2022. Fantastically Ordered Prompts and Where to Find Them: Overcoming Few-Shot Prompt Order Sensitivity. In Proceedings of the 60th Annual Meeting of the Association for Computational Linguistics (Volume 1: Long Papers), pages 8086–8098, Dublin, Ireland. Association for Computational Linguistics.

**Paper Topic And Main Contributions:**

This paper measures the causal reasoning abilities of GPT-3.5 and GPT-4. This paper states that evaluating ChatGPT in terms of causal reasoning is important, but not done by current work. To fill this gap, this paper measures 4 ChatGPT models on 5 benchmarks (classified into three types). Results show that these models are better at explaining causal relations instead of classifying them. Analysis shows that these models tend to hallucinate (supported with possible causes), and their performance is sensitive to prompts and the features (like numbers and distances) of causal events in a sentence.

**Questions For The Authors:**

A. According to Table 4, “4 pos + 48 neg” does lead to better performance (76.2 to 68.9). Does this mean “overall accuracy did not improve” (line 331)?

B. For Table 2, how do you interpret “AVG-BLEU” scores?

**Reasons To Accept:**

1. Given the popularity of ChatGPT, the topic has practical and timely significance.
2. This paper presents comprehensive experiments and analysis to support main claims.

**Reasons To Reject:**

1. This paper does not present sufficient novel findings. Hallucination in LLM has been discussed in [1] and [2]; sensitivity to prompt is mentioned in [3] and [4].
2. For the CEG task, the baselines are too old to be effective. There are more recent baselines like FLAN-T5 and LLaMA.
3. For analysis of in-context learning (sec 5.1), the order of few-shot demonstrations should be discussed, as it usually influences the performance.
4. This paper states that increasing lexical distances of events lead to worse causal reasoning performance (line 96-98, line 482-483). This is not supported by section 5.4 (line 405-408). In contrast, most models perform better as the distance increases (Figure 3).

**Reproducibility:**

3: Could reproduce the results with some difficulty. The settings of parameters are underspecified or subjectively determined; the training/evaluation data are not widely available.

**Reviewer Confidence:**

3: Pretty sure, but there's a chance I missed something. Although I have a good feel for this area in general, I did not carefully check the paper's details, e.g., the math, experimental design, or novelty.

**Typos Grammar Style And Presentation Improvements:**

Typo: line 423, “that” to “than”; line 303, “that” to “than”.

Presentation: it might be better if this paper interprets these terms before using them: “causal interpreter” (explained in sec 4.1.2), “hallucination” (explained in sec 4.1),  and “lexical distance” (sec 5.4).

---

> ### Author Rebuttal · Authors · 2023-08-29
>
> ## 1. Hallucination in LLM has been discussed in [1] and [2].
>
> > [1] Bang, Y., Cahyawijaya, S., Lee, N., Dai, W., Su, D., Wilie, B., ... & Fung, P. (2023). A multitask, multilingual, multimodal evaluation of chatgpt on reasoning, hallucination, and interactivity. arXiv preprint arXiv:2302.04023.
> >
> > [2] Manakul, P., Liusie, A., & Gales, M. J. (2023). Selfcheckgpt: Zero-resource black-box hallucination detection for generative large language models. arXiv preprint arXiv:2303.08896.
>
> Thanks for the review. Previous works [1] and [2] discussed the issue of illusions. However, our findings are not contained by [1] and [2]. Furthermore, we correct the misconception in [1], which believes that ChatGPT is good at causal reasoning.
>
> **For [1]:** [1] emphasizes the diversity of the evaluated tasks, while our work focuses on the comprehensive evaluation of causal reasoning tasks. For causal reasoning, [1] only tested 30 examples from the e-CARE [3] dataset and incorrectly claimed that ChatGPT achieved good performance, which our experiments demonstrate not to be the case. Besides, we identified a characteristic causal illusion issue in ChatGPT, where it tends to fabricate causal relationships between events. We further analyzed the possible reasons for such illusion and tested ChatGPT's performance in settings like in-context learning and chain-of-thought. Additionally, [1] lacks the experimental results of GPT-4.
>
> > [3] Du L, Ding X, Xiong K, et al. e-CARE: a New Dataset for Exploring Explainable Causal Reasoning[C]//Proceedings of the 60th Annual Meeting of the Association for Computational Linguistics (Volume 1: Long Papers). 2022: 432-446.
>
> **For [2]:** There is no mention of any causal reasoning task in [2], and [2] focuses on offering a general solution to the illusion of ChatGPT rather than evaluating the causal reasoning capabilities. Besides, The hallucination issue of ChatGPT is only one aspect analyzed in our work. Therefore, compared to ours, [2] belongs to a different line of research.
>
> **Besides:** In addition to experiments and findings related to illusions, we also analyzed various factors influencing ChatGPT's causal reasoning ability, including the ways of expressing causality in prompts, the number of events in the sentence, the lexical distance between events, the type of causality (implicit or explicit), and whether an open-ended question format is utilized.
>
>
>
> ## 2. Sensitivity to prompt is mentioned in [3] and [4].
>
> Clearly, sensitivity to the prompt is a well-known characteristic of the ChatGPT, but it is necessary for a comprehensive evaluation to verify and report it in the context of causal reasoning tasks. Regarding prompt variations, we not only reported ChatGPT's sensitivity but also conducted a detailed analysis of the effects of different causal prompts, including the ways of expressing causality in prompts and whether an open-ended question format is utilized.
>
> Sensitivity to the prompt is just one of our secondary contributions. Our work evaluated four causal reasoning task formats, five datasets, and four versions of ChatGPT, including GPT-4. Our other contributions can be summarized as follows:
>
> 1. We empirically demonstrated ChatGPT's performance in causal reasoning.
> 2. We summarized the characteristic of illusions in ChatGPT's causal reasoning.
> 3. We analyzed potential reasons for such causal illusion issues.
> 4. We investigated the impact of techniques like ICL and CoT on causal illusions of ChatGPT.
> 5. We studied the influence of various prompt-agnostic factors, including event density, event spacing, and causal types.
>
>
>
> ## 3. For the CEG task, the recent LLM baseline (e.g., LLaMA and FLAN-T5) should be added.
>
> Thank you for constructive suggestions. Our work focused on ChatGPT, so there was no further discussion on the performance of other types of LLMs in causal reasoning. Instead, we used fine-tuned small PLMs as baselines to contrast the differences between zero-shot LLM and fine-tuned small PLMs. We will add the following CEG experiments with fine-tuned LLaMA 7B and FLAN-T5 11B in the revised version.
>
> | Method           | AVG-BLEU | ROUGE-l | Human Evaluation |
> | :--------------- | :------: | :-----: | :--------------: |
> | GRU-Seq2Seq      |   18.7   |  21.3   |        0         |
> | GPT2             |   32.0   |  31.5   |       20.0       |
> | **LLaMA 7B**     |   40.3   |  37.1   |       63.0       |
> | **FLAN-T5 11B**  |   38.5   |  42.8   |       66.0       |
> | text-davinci-003 |   10.1   |  37.5   |       83.0       |
> | gpt-3.5-turbo    |   7.3    |  40.3   |       82.0       |
> | gpt-4            |   6.5    |  39.8   |       85.0       |
>
> For LLaMA 7B and FLAN-T5 11B, we utilize LoRa to perform parameter-efficient fine-tuning on the e-CARE dataset. We find that fine-tuned LLaMA and FLAN-T5 achieved significant better performance compared to GRU-Seq2Seq and GPT2. This is mainly due to the increased model scale enhancing the model's capability.
>
> In manual evaluation, we find that the explanations generated by LLaMA and FLAN-T5 are highly correlated with the input events. However, the explanations might be mere repetitions of the input events or provide relevant but uninformative descriptions that cannot be used for explanation.
>
> Besides, compared to ChatGPT, the explanations provided by these two fine-tuned LLMs are noticeably shorter, as the gold explanations provided by e-CARE are very concise. However, ChatGPT excels in providing more comprehensive and detailed explanations in the zero-shot setting.
>
> For AVG-BLEU, it is a text generation metric similar to "precision" in text classification tasks. Therefore, ChatGPT received a lower AVG-BLEU score because it generated detailed explanations that were not included in the annotations from e-CARE.
>
> For ROUGE-l, it is a text generation metric similar to "recall" in text classification tasks. It's worth noting that fine-tuned LLMs and ChatGPT achieve similar ROUGE-l scores, indicating that they all have some gap from the gold explanations in e-CARE. This is because that the fine-tuned LLaMA and FLAN-T5 may generate less informative explanations compared to e-CARE, e.g., mere repetitions of the input events. However, ChatGPT may offer valuable explanations for the input causal event pairs, but from different perspectives or in distinct syntactic forms compared to the gold explanations in e-CARE.
>
>
>
>
>
>
>
> ## 4. The impact of the order of few-shot demonstrations.
>
> In the early experiments, we conducted multiple random tests on the ICL performance of the model. We shuffled the order of few-shot demonstrations, but the performance did not show significant changes. Therefore, there was no further discussion on the order. We will add the following discussion on order of few-shot demonstrations into the revised version.
>
> For the few-shot setting "top 4 similar" in Sec 5.1, we analyze the model's performance under different demonstration orders:
>
> 1. **Causal First**: from causal demonstrations to non-causal demonstrations.
> 2. **Non-Causal FIrst**: from non-causal demonstrations to causal demonstrations.
> 3. **Random**, conducting three times experiments under random demonstration orders.
>
> "Zero-shot" indicates the performance under the zero-shot setting.
>
> | Method           | Pos Acc | Neg Acc | Full Acc |
> | ---------------- | :-----: | :-----: | :------: |
> | Zero-shot | 74.4    | 67.7    | 68.9     |
> | Causal FIrst     | 97.1    | 23.4    | 35.1     |
> | Non-Causal FIrst | 92.8    | 29.0    | 39.2     |
> | random1          | 95.6    | 28.0    | 38.8     |
> | random2          | 95.7    | 27.8    | 38.6     |
> | random3          | 95.7    | 27.4    | 38.3     |
>
> We find that Causal First is more inclined to classify event pairs as causal compared to Non-Causal First. This might be because that the demonstrations located earlier have a stronger impact on the model.
>
> In addition, despite different orders, all of these few-shot settings make the model more inclined to classify event pairs as causal compared to "zero-shot".
>
>
>
> ## 5. Lexical distances in section 5.4
>
> As explained in lines 399-408, with the increase in the distance between events: (1) the proportion of non-causal events among all event pairs will rise; (2) the correlation between events decreases, causing the model to be more inclined to assign event pairs as non-causal. The combination of these two effects results in Full acc becoming an imbalanced metric, reflecting only the increase in accuracy of non-causal pairs but not reflecting the decrease in accuracy of causal pairs.
>
> To further alleviate the confusion caused by the Full acc metric, we will add the F1 score as a more comprehensive metric in the revised version. The following table shows the changes in the F1 score of ChatGPTs across different distances between events. An outlier is the F1 score of GPT-4 at the interval [25,30). This is due to the fact that out of 1000 test instances for GPT-4, there are only 35 examples within the interval [25,30), leading to more random performance. However, all other results demonstrate that ChatGPTs' performance decreases as the event distance increases.
>
> | Model            | [1,5) | [5,10) | [10,15) | [15,20) | [20,25) | [25,30) | [30,+) |
> | :--------------- | :---: | :----: | :-----: | :-----: | :-----: | :-----: | :----: |
> | text-davinci-002 | 44.6  |  44.1  |  39.7   |  36.5   |  32.4   |  30.2   |  29.0  |
> | text-davinci-003 | 52.3  |  49.8  |  47.3   |  44.7   |  42.7   |  41.9   |  35.5  |
> | gpt-3.5-turbo    | 49.0  |  46.9  |  42.0   |  39.2   |  34.7   |  39.9   |  28.6  |
> | gpt-4            | 51.8  |  48.9  |  35.8   |  41.0   |  34.5   |  62.5   |  25.0  |
>
>
>
> ## 6. Performance changes of "4 pos + 48 neg" in Table 4.
>
> Thank you for careful review. This was an error in writing line 331, "4 pos + 48 neg" is one of our 29 few-shot experimental results, the overall accuracy of "4 pos + 48 neg" **does** improve. The error did not occur in Table 4, where the reported experimental results are reliable. Thank you for pointing out this error. We will fix it and clarify the confusion in the revised versions.
>
> In the analysis of ICL and CoT (Table 4, 5, and 6), it can be observed that several methods, including "4 pos + 48 neg", outperform the zero-shot method in terms of Full acc. However, a common trend among these methods is that they enhance Pos acc to improve Neg acc, or vice versa. Besides, the same method performs inconsistently across different versions of ChatGPT.
>
> We believe that the improvement of overall performance in causal reasoning ability should not show a compromise between identifying causal and non-causal pairs. Such behavior, including "4 pos + 48 neg", would merely be a case of robbing Peter to pay Paul, rather than a substantial improvement in the overall performance of causal reasoning.
>
>
>
>
>
> ## 7. For Table 2, how do you interpret “AVG-BLEU” scores?
>
> **What is the AVG-BLEU that we used:**
>
> As mentioned in line 172-183, we conducted experiments using the e-CARE dataset and employed the AVG-BLEU metric as described in the e-CARE paper for evaluation. Specifically, e-CARE first calculates the BLEU score for each test instance and then compute the average of all calculated BLEU scores to obtain the AVG-BLEU for the entire test dataset. We utilized the open-source code provided by e-CARE to compute the AVG-BLEU.
>
> **Explanation of the experimental results for AVG-BLEU in Table 2：**
>
> **AVG-BLEU** is an evaluation metric similar to precision. Roughly speaking, it calculates the ratio of the words appearing in both the generated output and the gold response text to the length of the generated output. As mentioned in line 302-307, ChatGPT tends to generate  more complete and detailed explanations. However, the explanations annotated in e-CARE are shorter and more concise, and the AVG-BLEU score will only consider the output part annotated in e-CARE as correct, treating all other output part as incorrect. Therefore, ChatGPT obtained relatively low AVG-BLEU scores. In addition, compared to text-davinci-003, both gpt-3.5-turbo and gpt-4 have a lower AVG-BLEU. This is because they have further trained by RLHF and tend to generate longer responses.
>
> On the contrary, **ROUGE-l** is a metric similar to recall, focusing on whether the model's responses contain the annotated gold explanations. As a result, ChatGPT exhibits contrasting trends in ROUGE-l and AVG-BLEU.
>
>
>
> ## 8. Suggestions for improving the presentation.
>
> Thank you for valuable suggestion. We will include explanations for the terms "causal interpreter", "hallucination", and "lexical distance" in the revised versions.

---

### Official Review · Reviewer_SCUQ · 2023-08-03

**Typos Grammar Style And Presentation Improvements:** line 230 event -> even
**Soundness:** 4

**Excitement:**

3: Ambivalent: It has merits (e.g., it reports state-of-the-art results, the idea is nice), but there are key weaknesses (e.g., it describes incremental work), and it can significantly benefit from another round of revision. However, I won't object to accepting it if my co-reviewers champion it.

**Paper Topic And Main Contributions:**

This study conducts a comprehensive evaluation of the ChatGPT family of models on their ability of four causal reasoning tasks from five datasets. The evaluation results suggest the current limitation of ChatGPT on this task. The findings would be insight for similar evaluations on LLMs.

**Questions For The Authors:**

a. As mentioned in the "Key takeaways" - ChatGPT is sensitive to the words used to express the causal concept in the prompt- could the authors elaborate more on the exact prompts used under the zero-shot setting for each of the tasks? I wonder if the questions asked in the prompt would affect the performance of the models.

b. For model performance evaluation, could the authors elaborate more on how the responses of ChatGPT were mapped to the desired format that could be compared to the ground truth of different datasets? For example, previous studies have suggested that the direct inferencing results from ChatGPT models after adding CoT may contain extra argument and elaboration. It would be helpful if the authors could explain a bit more about their methods for processing the response.

c. Table 1 and Table 2 provide the fine-grained results for ChatGPT on the pos, neg classes. How about the ones for the pre-trained baseline models? Would the fine-tuned models also do better at identifying causal event pairs but not in non-causal event pairs, or vice versa?

d. For the "top k similar", could the authors elaborate more on whether the k examples are within one class (neg or positive) or from both classes? How might it influence the evaluation results?

**Reasons To Accept:**

Causal reasoning is an important task in the NLP community. Having a thorough evaluation of the available LLMs like chatGPT on this task would be informative for the community and also helps with the future improvement regarding the performance of the LLMs.

Compared to existing evaluations that normally took a few hundred examples for evaluation, the current study covers more datasets and tasks specifically for causal reasoning tasks.

Besides standard evaluation metrics (precision, f1, accuracy, etc.), the study also provides a detailed discussion and comparison of the model performance.

**Reasons To Reject:**

There is not much algorithm breakthrough.

**Reproducibility:**

4: Could mostly reproduce the results, but there may be some variation because of sample variance or minor variations in their interpretation of the protocol or method.

**Reviewer Confidence:**

3: Pretty sure, but there's a chance I missed something. Although I have a good feel for this area in general, I did not carefully check the paper's details, e.g., the math, experimental design, or novelty.

---

> ### Author Rebuttal · Authors · 2023-08-29
>
> ## 1. The forms of prompts.
>
> **As we mentioned in Sec 3.2 Experiment Setting (lines 186-188), we illustrated the exact prompts used under the zero-shot setting for each of the tasks in Figure 1.**
>
> **The prompt of the ECI task:**
>
> Input: \<input sentence\>
>
> Question: is there a causal relationship between “\<event1\>”and “\<event2\>” ?
>
> Answer:
>
> **The prompt of the multiple choice CD task:**
>
> *(This task involves samples that ask for selecting the result of the input event. For such samples, we modify the word “cause” in the question to the word “result”.)*
>
> Input Event: \<input event\>
>
> Question: Please select the cause of the input event from the following options.
>
> Option 1: \<option1\>
>
> Option 2: \<option2\>
>
> Answer:
>
> **The prompt of the Binary Classification CD task:**
>
> Event A: \<input event1\>
>
> Event B: \<input event2\>
>
> Question: is there a causal relationship between Event A and Event B ?
>
> Answer:
>
> **The prompt of the CEG task:**
>
> Cause: \<input cause\>
>
> Effect: \<input effect\>
>
> Question: why the cause can lead to the effect?
>
> Answer:
>
>
>
> ## 2. The impact of different questions in prompts.
>
> We discuss if the questions asked in the prompt would affect the performance of ChatGPT in Sec 5.3 and Sec 5.7.
>
> In Sec 5.3, we analyze the performance of ChatGPT with prompts that express the causal concept in different ways. We find that:
>
> 1. The prompt based on the counterfactual causality view exacerbates the causal hallucination issue.
> 2. Using the phrase "one-step" to restrict the captured causal relationships can improve performance on the Neg class, but significantly reduces performance on the Pos class.
> 3. The performance differences of various causal cue words in prompts, such as "lead to" and "result in", are quite significant. This further indicates that it is challenging to accurately convey what causality means to ChatGPT through prompts.
>
> In Sec 5.7, we analyze open-ended questions in the prompt, directly asking the ChatGPT to generate pairs of causal events in the text, rather than responding "yes" or "no" to specified target event pairs. We find that the open-ended prompts decrease the performance of ChatGPT. This is mainly because open-ended prompts require ChatGPT to jointly perform the event extraction and the causal reasoning task. However, previous studies show that ChatGPT is not good at extracting events.
>
>
>
> ## 3. How the responses of ChatGPT were mapped to the desired format?
>
> Our goal is to reflect the actual performance of ChatGPT, avoiding inaccurate evaluation results stemming from wrong automated mapping rules.
>
> **For the CEG task:**
>
> no further text map is necessary; we directly use the responses of ChatGPT as its final prediction.
>
> **For the ECI task**:
>
> Under the zero-shot setting, we found that out of 138,166 instances, only 96 responses do not begin with "yes" or "no". Therefore, responses starting with "yes" will be assigned the label 1, while other cases will be assigned the label 0.
>
> Under the ICL and CoT settings, we construct demonstrations ending with the labels "yes" or "no". Therefore, responses from ChatGPT ending with "yes" will be mapped to label 1, while others will be mapped to label 0.
>
> For the prompts discussed in Section 5.3, these prompts are directional. Therefore, for a pair of events, we reverse their order to obtain two responses from ChatGPT. If at least one of the two responses starts with "yes", it is assigned the label 1, while other cases will be categorized as label 0.
>
> For the open-ended prompt discussed in Section 5.7, we added descriptions in the prompt about the output format, allowing for easy parsing into causal event pairs.
>
> **For the CD task:**
>
> Most responses begin with phrases "option1", "option2", "yes", or "no", which can be easily mapped to labels. For other response formats, we manually assigned labels for each different format of response, and the mapping results are recorded in the supplementary file "CD_and_CEG_compute_score.py".
>
>
>
> ## 4. Would the fine-tuned models also do better at identifying causal event pairs but not in non-causal event pairs, or vice versa?
>
> Thank you for constructive suggestions! As we primarily focus on ChatGPT, and there is no available previous performance reports for Pos and Neg acc with fine-tuned models, we did not provide them on Table 1 and Table 2. The following table presents these results, would be in the revised version.
>
> | For Table 1 (ECI) |         |   ESC   |          |         |   CTB   |          |         | MAVEN-ERE |          |
> | :---------------- | ------- | :-----: | :------: | :-----: | :-----: | :------: | :-----: | :-------: | :------: |
> | **Methods**       | **Pos** | **Neg** | **Full** | **Pos** | **Neg** | **Full** | **Pos** |  **Neg**  | **Full** |
> | BERT-Base         | 59.5    |  83.6   |   79.7   |  46.4   |  87.8   |   86.2   |  75.7   |   86.9    |   85.1   |
> | RoBERTa-Base      | 63.8    |  82.8   |   79.8   |  62.3   |  86.4   |   85.5   |  76.6   |   87.1    |   85.4   |
>
> **In the ECI task:** it can be observed that the fine-tuned models do better at identifying non-causal event pairs. This is because there are much more negative examples than positive examples in the ECI dataset, and the fine-tuned models have learned such data distribution.
>
> | For Table 2 (CD) |         | e-CARE  |          |         |  COPA   |          |
> | ---------------- | :-----: | :-----: | :------: | :-----: | :-----: | :------: |
> | **Methods**      | **Pos** | **Neg** | **Full** | **Pos** | **Neg** | **Full** |
> | BERT-Base        |  65.7   |  95.2   |   65.0   |  59.5   |  88.9   |   54.5   |
> | RoBERTa-Base     |  64.6   |  74.6   |   64.5   |  66.0   |  69.0   |   63.5   |
>
> **In the CD task:** we find that the fine-tuned models do better at identifying non-causal event pairs. This may be because, as mentioned in lines 268-272, the non-causal pairs in the e-CARE and COPA datasets are manually annotated given a input event, and they have a simple structure and weak correlation with the input events, making them easier to identify.
>
> ## 5. The classes of demonstrations in "top k similar".
>
> I appreciate your valuable feedback. In the experiment, we did not constrain the class of the demonstrations to be Pos or Neg.
>
> Below, we analyze the impact of the classes of demonstrations on performance of "top k similar". For k=4, we first divide the dataset into five subsets, each containing instances that only use 0, 1, 2, 3, or 4 Pos demonstrations in their top 4 similar demonstrations, respectively. Then, we present the performance of the "top 4 similar" on these five subsets. "Proportion of Pos" is the proportion (%) of Pos instances in the corresponding subset.
>
> | Used Demonstrations | Pos Acc | Neg Acc | Full Acc | Proportion of Pos |
> | ------------------- | :-----: | :-----: | :------: | :---------------: |
> | 0 pos 4 neg         |  98.4   |  21.8   |   26.7   |        6.4        |
> | 1 pos 3 neg         |  88.8   |  27.5   |   39.1   |       18.9        |
> | 2 pos 2 neg         |  88.6   |  36.3   |   49.5   |       25.2        |
> | 3 pos 1 neg         |  89.5   |  47.1   |   66.9   |       46.6        |
> | 4 pos 0 neg         |  95.3   |  35.7   |   83.7   |       80.6        |
>
> We can observe that when using only Pos or Neg demonstrations, the model achieves a higher Pos acc. This might be because that including only one classes prevents the model from contrasting the meanings of different labels, thus potentially confusing the model's understanding of the task objectives.
>
> Additionally, when using both Pos and Neg demonstrations, there is a smaller change in Pos acc, while Neg acc increases as the num of Pos demonstrations rises. This might be because having more Pos demonstrations helps the model understand the situations that truly involve causality, thus avoiding misclassifying non-causal instances as causal.

---

### Official Review · Reviewer_MYMZ · 2023-08-04

**Soundness:** 4

**Excitement:**

3: Ambivalent: It has merits (e.g., it reports state-of-the-art results, the idea is nice), but there are key weaknesses (e.g., it describes incremental work), and it can significantly benefit from another round of revision. However, I won't object to accepting it if my co-reviewers champion it.

**Paper Topic And Main Contributions:**

This paper perform a comprehensive study to evaluate the causal reasoning abilities (on event causality identification, causal discovery and causal explanation generation tasks) of GPT models including text-davinci-002, text-davinci-003, chatgpt and gpt4. They find that ChatGPT models can not do the causal reasoning so well and have serious hallucination issues. They also show that the hallucinations might be due to the RLHF.


**Questions For The Authors:**

For the error cases, what are some common error types? Are there any patterns and are there some reasons? Does different model show similar distributions?

**Reasons To Accept:**

1. The paper comprehensively evaluate the causal reasoning abilities of gpt models together with several insights summarized from experiments.
2. The analysis is interesting and might lead to future improvements for causal reasoning.


**Reasons To Reject:**

1. It would be interesting to show the error analysis from the results. For example, what are some common types for the hallucination?
2. The authors claim that the hallucination might be from the RLHF. It might be great to have the some preliminary/toy analysis on that. For example, the same model before and after RLHF.

**Reproducibility:**

4: Could mostly reproduce the results, but there may be some variation because of sample variance or minor variations in their interpretation of the protocol or method.

**Reviewer Confidence:**

5: Positive that my evaluation is correct. I read the paper very carefully and I am very familiar with related work.

---

> ### Author Rebuttal · Authors · 2023-08-29
>
> ## 1. Error Analysis
>
> Thanks for insightful suggestions. We will incorporate error analysis into the revised version.
>
> We analyze the types of errors by observing the reasoning chains generated in the CoT setting. Common error types include:
>
> 1. Fabricating additional fake conditions to establish causality, even if these conditions are not described or are incorrect in the input sentence.
> 2. Misunderstanding basic event relationships such as sub-events and temporal relationships between events.
> 3. Failing to accurately identify which two events the causal question is referring to.
> 4. Introducing incorrect commonsense knowledge.
>
> We believe that causal reasoning is a comprehensive skill that requires commonsense knowledge, as well as the ability to understand basic event relationships, and to perform logical reasoning based on information. However, ChatGPT is still not entirely reliable in these aspects, leading to the accumulation of errors. On the other hand, ChatGPT has encountered numerous causal event pairs in pre-training data, enabling it to associate any event pair with a potentially causal context. However, this context might not align with the input.
>
> We randomly selected 100 instances of errors for each model and manually annotated them to analyze the types of errors among different models. Compared to text-davinci-003, which is not fine-tuned on dialog data, gpt-3.5-turbo and gpt-4 show a clearer tendency to fabricate additional fake conditions to establish causality. This could be due to dialog data guiding them to produce longer and more divergent responses, which deviate from the context provided in the original input. Additionally, gpt-4 introduces fewer incorrect commonsense knowledge, as it has a better grasp of knowledge.
>
> **Table. Distribution of Error Types across Various Models on 100 Error nstances.**
>
> | Error Type                          | text-davinci-003 | gpt-3.5-turbo | gpt-4 |
> | :---------------------------------- | :--------------: | :-----------: | :---: |
> | Fake conditions                     |        31        |      39       |  45   |
> | Incorrect basic event relationships |        19        |      17       |  16   |
> | Wrong target event localization     |        15        |      11       |  13   |
> | Incorrect commonsense knowledge     |        12        |      15       |   9   |
> | Other                               |        23        |      18       |  17   |
>
>
>
> ## 2. Further Analyze on RLHF
>
> text-davinci-002 was trained without RLHF. As described in lines 191-203, text-davinci-003, gpt-3.5-turbo, and gpt-4 are the results of further RLHF applied to text-davinci-002. By comparing text-davinci-002 (before RLHF) to other three models (after RLHF), we concluded in lines 280-293 that RLHF exacerbates the issue of causal hallucination in ChatGPT.
>
> Our work focused on ChatGPT, and due to OpenAI's decision not to open-source it, we lack access to specific details about the data and experimental setup for ChatGPT's RLHF. As a result, we did not provide further analysis about the RLHF of ChatGPT.
>
> As an alternative approach, we will add RLHF analysis based on the Anthropic RLHF dataset [1], which is an open-source RLHF dataset. Some of our preliminary findings include:
>
> 1. Among questions about "Why", only 10.29% include the word "not". This indicates that the majority of trianing data in the dataset guide the LLM about causality rather than non-causality.
> 2. In the annotated gold responses, the frequency of the word "yes" is approximately twice that of the word "no". This could potentially increase the likelihood of the LLM producing the positive label rather than the negative label.
>
> Both of these characteristics in the RLHF data could potentially exacerbate ChatGPT's causal hallucination, leading it to assume causal relationships between events, regardless of whether those relationships actually exist.
>
> A further analysis of why RLHF exacerbates the issue of causal hallucination requires a controlled variable analysis of the RLHF data, training iterations, base model, etc. This necessitates the training of LLM under various RLHF settings, which is time-consuming. Due to time constraints, we could not provide this analysis during the rebuttal period, but it will be included in the revised version. For instance, whether balancing the RLHF dataset can alleviate the causal hallucination issue in LLM.
>
> > [1] Bai Y, Jones A, Ndousse K, et al. Training a helpful and harmless assistant with reinforcement learning from human feedback[J]. arXiv preprint arXiv:2204.05862, 2022.

---

### Meta-Review · Area_Chair_E4uK · 2023-09-19

**Recommendation:** 3

**Metareview:**

The reviewers generally agree that the paper provides a comprehensive evaluation of the causal reasoning abilities of GPT models, offering valuable insights for future improvements. However, they also point out that the paper lacks novelty, with some findings already discussed in previous studies. They suggest that the authors could have used more recent baselines for the Causal Explanation Generation task and discussed the order of few-shot demonstrations. The reviewers also note inconsistencies in the paper's claims and results, and suggest clarifications on certain terms and results. The authors' rebuttals address most of the concerns raised, providing detailed explanations and promising to include additional analyses in the revised version.

---

### Decision · Program_Chairs · 2023-10-07

**Decision:**

Accept-Findings

**Comment:**

The reviewers generally agree that the paper provides a comprehensive evaluation of the causal reasoning abilities of GPT models, offering valuable insights for future improvements. However, they also point out that the paper lacks novelty, with some findings already discussed in previous studies. They suggest that the authors could have used more recent baselines for the Causal Explanation Generation task and discussed the order of few-shot demonstrations. The reviewers also note inconsistencies in the paper's claims and results, and suggest clarifications on certain terms and results. The authors' rebuttals address most of the concerns raised, providing detailed explanations and promising to include additional analyses in the revised version.